# Understanding *Ustilago maydis* Infection of Multiple Maize Organs

**DOI:** 10.3390/jof7010008

**Published:** 2020-12-27

**Authors:** Alex C Ferris, Virginia Walbot

**Affiliations:** 1Department of Bioengineering, Stanford University, Stanford, CA 94305, USA; 2Department of Biology, Stanford University, Stanford, CA 94305, USA; walbot@stanford.edu

**Keywords:** *Ustilago maydis*, effectors, anthers, male sterile maize

## Abstract

*Ustilago maydis* is a smut fungus that infects all aerial maize organs, namely, seedling leaves, tassels, and ears. In all organs, tumors are formed by inducing hypertrophy and hyperplasia in actively dividing cells; however, the vast differences in cell types and developmental stages for different parts of the plant requires that *U. maydis* have both general and organ-specific strategies for infecting maize. In this review, we summarize how the maize–*U. maydis* interaction can be studied using mutant *U. maydis* strains to better understand how individual effectors contribute to this interaction, either through general or specific expression in a cell type, tissue, or organ. We also examine how male sterile maize mutants that do not support tumor formation can be used to explore key features of the maize anthers that are required for successful infection. Finally, we discuss key unanswered questions about the maize–*U. maydis* interaction and how new technologies can potentially be used to answer them.

## 1. Introduction

*Ustilago maydis* is a biotrophic fungus that causes tumors in all aerial maize organs. After spores germinate on the epidermis, in nature two compatible strains fuse to form a dikaryotic filament that can infect maize cells; however, the lab-derived SG200 strain is solopathogenic and utilized in almost all experiments [1,2]. *U. maydis* initially grows along the epidermis of all organs before penetrating between epidermal cells to reach and then spread throughout the subepidermal cells. It forms a biotrophic interface (close juxtaposition of fungal cell wall with the plant cell wall or plasma membrane) with the target interior cells, which invaginate their plasma membranes to accommodate branching fungal hyphae [1]. After approximately 3 to 5 days, there is substantial growth (hypertrophy) and excess host cell division (hyperplasia), the initial steps of tumor formation [3,4]. By convention, botanists define organs as an assemblage of tissues, and tissues are defined as containing one or more cell types that constitute a functional unit, i.e., photosynthetic leaf tissue contains mesophyll and bundle sheath cell types. *U. maydis* infects leaf, stem, and reproductive organs (ears, tassels) of maize. The differential interaction of the fungus and epidermal compared to interior cell types demonstrates tissue-specific interactions between host and pathogen in all organs examined.

### 1.1. Leaf Characteristics

The tissues in leaf and anther are distinct in terms of both cell composition and relative developmental stage at the time of inoculation, although tumors develop in approximately the same amount of time. The outer layer of leaves is composed of epidermal cells—pavement, bulliform, subsidiary, hair, and guard cell types. In the center of the leaf there is vasculature (sieve tube, companion, xylem, and parenchyma cell types) surrounded by bundle sheath cells and mesophyll cells. (Figure 1A). Fungi inoculated onto seedling and subsequent leaves grow towards the vasculature, a source of nutrition, and basipetally towards the youngest cells of the blade, a meristematic zone with actively dividing cells near the joint between the blade and sheath of each leaf. Fully mature leaves are more recalcitrant to infection than leaves that are still growing. 

### 1.2. Reproductive Organ Characteristics

Maize tassels contain hundreds of paired spikelets covered by the leaf-like glumes that each contain two florets (including the palea, lemma, three stamens, a pair of lodicules) [6]. Stamens contain two distinct organs—the filament (epidermis, connective tissue, vasculature) subtending an anther. Anthers are the most rapidly growing spikelet organ and will ultimately constitute most of the floret mass. Anthers contain the continuation of the vasculature and connective tissue from the filament and four pollen-producing lobes. During their period of peak cell proliferation, each tassel organ can be converted to tumorous growth by *U. maydis* infection. For example, successful *U. maydis* infection occurs in pre-meiotic anthers, starting as early as the anther primordium stage and continuing through the period of rapid cell proliferation when anther lobes achieve a dartboard architecture with five layers of cells: epidermis, endothecium, middle layer, tapetum, and archesporial cells (Figure 1B) [6]. The first four tissues of anther cells are strictly somatic and only one cell thick. This period of anther development requires approximately 7 days. Initially, anther primordia of 50–150 µm consist of epidermis (layer 1 of the floral meristem) and layer 2-derived cells, with archesporial cells beginning to differentiate around 170 µm [6]. Somatic cell patterning begins around 240 µm as endothecial and secondary parietal cells are specified after periclinal division of the layer 2-derived cells, and finally the tapetal and middle layer cells differentiate after periclinal division of secondary parietal cells during the 600–700 µm stage [6]. The vegetative cells proliferate rapidly through the 1 mm stage. On the basis of the well-established timeline of events in maize anthers, this means that if 400 µm anthers are infected with *U. maydis*, the fungus will reach the subepidermal cells when the anthers are approximately 700–900 µm, which is after formation of all five cell types and after the earlier acting male sterile mutants affect anther development.

The maize cob, equivalent to the tassel organ, contains the same diverse tissues as the male inflorescence. In both the cob and tassel, florets initiate with perfect flowers (glumes, palea, lemma, lodicules, stamens, carpel); however, in the tassel, the carpels abort, resulting in male-only flowers. In the cob, the stamens as well as the lower florets abort, resulting in a single viable carpel per spikelet. Despite the considerable reduction in rapidly growing tissue in the ear, in the field, ear tumors from the conversion of individual kernels (equivalent to an anther) are a striking feature of *U. maydis* infection. To date, few experiments have addressed tumor progression in ear tissues. It is unknown whether the same (or homologous or paralogous) host genes contribute to tumor progression or whether *U. maydis* uses a different suite of effectors to colonize ear cell types compared to tassels.

### 1.3. Infection and Tumor Progression

The specific details of *U. maydis* infection vary between seedling and adult leaves as well as between leaves and anthers. In *U. maydis* seedling leaf infections, an extensive biotrophic interface forms around 4 days post-infection (dpi) when the hyphae have often colonized meristematic tissue at the blade base and the bundle sheath and mesophyll cells start to be converted into tumor cells in the fully differentiated distal blade tissue (see Figure 1A) [4,7,8]. At this point, the epidermal and mesophyll cells begin to enlarge, eventually doubling and tripling in size, respectively, by 13 dpi [4]. In anthers, *U. maydis* reaches the subepidermal cells at approximately 3 dpi, and then extra periclinal divisions are observed in all the somatic tissues [3]. Although middle layer cells typically die early during anther development, in infected anthers, they exhibit both hyperplasia and hypertrophy and are the most obvious site of substantial growth [3]. By 7–10 dpi, the surface of infected tassel organs including anthers are distorted and enlarged, and by 15 dpi, mature tumors begin to split open and release diploid teliospores [9].

## 2. Exploiting *U. maydis* Mutants to Define Requirements for Tumor Formation

Much of the success or failure of *U. maydis* infection depends on effectors, secreted proteins that mediate the fungal interaction with maize. There are two broad classes of effectors: core effectors are secreted by the fungus regardless of which organ is being infected and tissue-specific effectors are differentially expressed depending on the plant organ, tissue, or cell type infected. Another key classification recognizes intercellular effector proteins that remain outside the host cells and intracellular effectors that reach host cytoplasm. As a result of the high evolutionary pressure on effector genes, they can be computationally predicted by identifying non-conserved genes across related species, with this being particularly true for *U. maydis* because of its ability to infect all aerial parts of maize, unlike closely related smuts that are restricted to floral organs [10]. For example, Fly1 is a conserved fungal chitinase that is involved in cell separation in axenic culture, but the *U. maydis* homolog has co-evolved to additionally suppress the maize pathogen response [11]. More generally, comparison of the *U. maydis* and *Sporisorium reilianum* genomes led to the identification of 43 divergent gene clusters in *U. maydis*, which are highly enriched for putative secreted proteins, and 7 previously characterized effector gene clusters [10].

Effector genes are generally characterized by performing knockouts, with the impact tested by infecting maize seedling leaves or tassels, and then scoring any change in disease phenotype on the basis of visual symptom severity with typical classification being no symptoms, chlorosis, ligular swelling in leaves, small tumors, moderate tumors, or severe tumors [12]. Key tissue-specific effectors can be identified if a knockout strain only causes a change in disease phenotype in a single maize organ or tissue or by measuring gene expression in multiple organs if the effect on disease phenotype is more subtle [13]. Additional techniques such as confocal microscopy and metabolomics can then be used to characterize the mechanism of function in greater detail.

A recent example is the characterization of Sta1, a core effector that is secreted primarily at 2 dpi and that disrupts fungal cell wall structure [14]. A SG200Δsta1 strain had significantly reduced disease symptoms in seedling leaves, and confocal microscopy at 6 dpi showed that the mutant *U. maydis* had reduced colonization between vascular bundles and that both mesophyll cells and bundle sheath cell development were similar to mock infected plants. Protein expressed constitutively or later during the infection under the control of heterologous promoters was unable to complement the SG200Δsta1 mutant. Fluorescently tagged Sta1 protein localized to the fungal plasma membrane, and filament formation was disrupted as detected by growing *U. maydis* on plates containing Congo red. Filamentous SG200Δsta1 cells were more susceptible to ß-glucanase and chitinase treatments compared to SG200, suggesting that Sta1 plays a key role in the structure of the fungal cell wall early in infection.

Limited sensitivity is a major constraint to screening mutants by scoring changes in disease phenotype for individual genes. In addition to the inherent variability between plants and individual subjective ratings, redundancy in gene function will fail to define key effectors. For example, the 19A gene cluster is predicted to contain 24 effector genes and one reverse transcriptase [15]. Deletion of the entire cluster drastically reduces *U. maydis* impact on infected seedling leaves compared to SG200, as do deletions of the *tin1-1*–*tin1-5* interval, or *tin2* or *tin3* genes within the cluster that also statistically reduced the disease phenotype [15,16]. None of these knock outs reproduce the increased levels of biotin found in the entire cluster deletion mutant, and the assay was not sensitive enough to characterize deletions of individual *tin1* mutants within the *tin1-1–tin1-5* interval. The mechanism of action has only been identified for *tin2*, which decreases anthocyanin by interacting with ZmTTK1, a protein kinase that increases anthocyanin biosynthesis [17].

In addition to core effectors such as Sta1 and Tin2, some tissue-specific effectors have also been characterized. Transcriptomics data can be used to identify effector genes that are differentially expressed, many of which cause tissue-specific changes in disease phenotype [13]. The best characterized of these is See1, which was demonstrated to reactivate cell division in seedling leaves and to be required for tumor formation there. The *see1* gene is dispensable in normal anthers because cell proliferation is already very active [18]. In leaves, Matei et al. [4] found a differential response of the two C4 photosynthetic partner cell types—bundle sheath cells exhibit hyperplasia, while neighboring mesophyll cells exhibit hypertrophy. Furthermore, See1 is required for reactivation of cell division in bundle sheath cells, refining its role to a cell type-specific action. In their analysis of tumor distribution, Matei et al. [4] also established that lignification of primary veins during infection restricts fungal lateral spread within the leaf; in the maize *brown midrib1* mutant, lignification is reduced, and fungal hyphae readily traversed the primary veins. After laser microdissection of infected mesophyll and bundle sheath cells and their mis-differentiated states later in infection, this group established cell type-specific transcriptomes, and specifically showcased the large role *see1* plays in regulating cell cycle genes [19].

### Future Directions

Mutant characterization through transcriptomics: One way to achieve greater sensitivity for determining if a mutant impacts *U. maydis* virulence is to look for meaningful differences in gene expression. A more in-depth time course characterization of the normal progression of SG200 infection would establish a better baseline for what to expect. Another way to increase sensitivity is to use single cell RNA-seq (scRNA-seq) instead of bulk organ (such as an entire leaf or anther) or tissue sequencing. This would provide information from both infected and uninfected cells in the same organ as well as identifying responses in individual cell types and decreasing the amount of noise in complex organ data. Using transcriptomics as a screening approach also has the advantage of generating better information about which pathways are being impacted by a given mutation as well as at what point infection development is interrupted.Involvement of fungal effectors in disease progression: Thus far, characterization of the mechanism of action of most effector genes is understood at a superficial level, i.e., the specific maize process targeted. Furthermore, it is difficult to connect effector disruption to what is changing in maize cells more generally. For example, what roles do secreted fungal effectors play in host cell entry into the tumor pathway? Moreover, which specific fungal effectors are required in sequential steps of host cell redirection into the tumor pathway? Are there multiple tumor pathways, depending on cell type?

## 3. Exploiting Maize Mutants to Define Requirements for Tumor Formation

Historically, genetic analysis of the *U. maydis*–maize interaction has exploited fungal mutants to define steps in tumor formation, and this trend was reinforced and expanded after publication of the *U. maydis* genome [2]. Furthermore, nearly all assays of pathogenicity relied on scoring symptom severity on maize seedlings. No significant maize resistance genes have been identified [20], suggesting that all maize lines would share similar responses to *U. maydis*. In effect, maize has been treated as a uniform substrate, and only fungal genotypes were varied in a laboratory setting. In contrast, in the field, seedling symptoms are rare, and tumors are most often scored on the vegetative tissues in the reproductive ears and tassels [21].

Until recently, exploration of the fungal–plant interaction was examined exploiting microbiological tools, almost entirely focused on the steps in pathogen development and the roles of effector genes. A few historic papers established maize tissue responses by microscopy [22,23]. Walbot and Skibbe [9] reported that tumor formation after tassel infections mirrored the sequence of peak proliferation in specific organs. First tassel primordia were converted to tumors, then whole spikelets not yet containing floral organs, then floral sterile organs (palea, lemma), followed by upper floret anthers, then the lower floret anthers (these are about 1 day younger than the upper floret anthers), and finally the lodicules (late expanding petal equivalents). A key observation was that maize mutants with early developmental arrest did not form anther tumors but could form tumors in the other tassel organs and on adult leaves. For example, both *msca1* and *mac1* anthers cease growth shortly after the normal time when meiosis starts and failed to form tumors in anthers. The *ms26* mutant, which fails to sustain growth about a week after completion of meiosis, formed anther tumors normally. *U. maydis* required about 14 days from inoculation to spore release—*msca1* and *mac1* anthers grow for about 7 days after the inoculation and *ms26* anthers for about 15 days. These findings suggest that *U. maydis* requires maize organs capable of growth to complete its lifecycle. A third observation from this study was that methyl jasmonate and brassinosteriod hormone injections could cause localized zones of male sterility with early growth arrest, in effect chemical phenocopies of the *msca1* or *mac1* state. Tumors failed to form in such growth-arrested zones, confirming that maize growth arrest restricts *U. maydis*. Further confirming growth arrest as a contributor to tumor failure, the *spi1* mutant, which forms a highly reduced tassel reflecting auxin-mediated growth failure, had almost no tassel tumors accompanied by normal leaf tumors; *kn1* plants, which exhibit ectopic growths on leaves, formed larger than normal tumors on both leaves and anthers. Collectively, these results demonstrate that manipulation of maize growth potential has a profound impact on *U. maydis* tumor success.

The inferred role of maize development, the requirement for growth potential, and the differential responses of adult leaf and tassel organs established by Walbot and Skibbe [9] inspired a comparative molecular analysis of maize–*U. maydis* interaction in seedling leaves, adult leaves, and tassels [24]. Organ type, tissue composition, and cell type identities are obviously distinct in these three maize organs, well-established by prior EST expression and microarray analyses [25]. The hypothesis tested was that *U. maydis* would express distinct suites of genes depending on the organ infected. This hypothesis was confirmed by analyzing both mRNA and protein expression at 1, 3, and 9 dpi. Both the host and pathogen showed stereotyped gene expression patterns on the basis of organ identity and time after infection. This study established a new paradigm that *U. maydis* tailors its interaction with maize on the basis of host organ identity and led to the quest to find organ-specific effectors and to the question of whether tissue and cell type-specific interactions exist.

Li et al. [3] reported that within anthers, excessive cell proliferation was stimulated preferentially in the middle layer of anther lobes; usually this tissue senesces during meiosis, however, in infected anthers, it not only persists but expands by cell proliferation and cell expansion. This surprising observation indicated that conversion to a tumor pathway could involve a specific cell type and that it could be difficult to predict which cell types participate in tumor formation.

A final example of organ-specific gene expression involves sugar transporters. Fungal capture of host nutrients is essential for hyphal growth and fungal maturation, and thus it is no surprise that the pathogen causes profound changes in the expression of various types of host sugar transporters. Sosso et al. [26] identified 13 differentially and tissue-specifically expressed sugar transporters during *U. maydis* infection: three *ZmSUTs*, seven *ZmSTPs*, and three *ZmSWEETs*. Downregulation of *ZmSUT1* would be expected to block leaf sugar efflux, increasing the local sugar concentration available to the fungus, while upregulation of three plasma membrane SWEETs in infected areas would increase soluble, apoplastic sugar concentrations.

### Future Directions

1.Exploiting maize mutants: The success thus far in exploiting maize mutants to dissect tumorigenesis should inspire additional experiments with host plants impaired in specific processes. Table 1 lists a few examples that could be fruitful. More information about each mutant is available at MaizeGDB by typing in a gene name or a process in the search box and requesting all data to retrieve phenotypic descriptions, photographs, genetic analysis, gene models, and references. Seeds are available for worldwide distribution from the Maize Coop (with links at MaizeGDB). Many of the examples involve generating a chimeric maize organ with normal and mutant tissue side-by-side. This arrangement permits scoring “normal” maize–fungal interaction near mutant host tissue–fungal interaction outcome.

2.Applying new analytical methods: New technologies also provide new opportunities to explore host–pathogen activities. Thus far, cell type information has been obtained on cell groups recovered by laser capture microdissection, but now that scRNA-seq is established for maize [27], an obvious approach is to evaluate the diverse cell types of seedling leaves, adult leaves, ears, and tassels in both infected and uninfected organs. Comparison of infected to uninfected cells between plants should identify both fungal and maize gene expression changes associated with infection progression resulting in tumor formation in every impacted cell type. Comparison of infected and uninfected cells within an infected organ from the same plant should highlight cell autonomous changes in host cells, i.e., those reflecting infection by *U. maydis*, and should allow discovery of non-cell-autonomous host responses as infected host cells alter the development or physiology of neighbors or cells distal from the infection. Effectors secreted extracellularly into the biotrophic interaction zone have a limited ability to diffuse to other cells; however, effectors secreted intracellularly have been shown to diffuse into and prime a response in secondary cells. In addition to the changes caused by effectors directly, maize cells can signal to each other through plasmodesmata or by utilizing secreted molecules (ions, hormones, peptides) or altered plasma membrane properties (lipid composition or modifications of existing lipids) that influence tissue or organ physiology or development. Alterations in host cell physiology during infection could thus be expected to cause non-cell-autonomous events, and it will be very interesting to determine the spatiotemporal distribution of this class of responses.

Van der Linde et al. [28,29] built a new tool, the Trojan horse, which engineered *U. maydis* to secrete maize proteins in addition to fungal effectors. This approach could be used to query the impact of maize proteins that alter cell-to-cell communication by utilizing non-pathogenic fungal strains to deliver presumptive host signaling factors.

3.Proteomics: Quantitative proteomics could contribute substantially to our understanding of host–pathogen interaction. mRNA analysis through RNA-seq is only a proxy for the “real actors” in host cell responses, the protein complement. In addition to improvements in mass spectrometry—both in the range of protein types that can be analyzed and miniaturization of sample size—that will be discovery tools, proximity labeling such as Turbo-ID can establish specific fungal effector–maize protein interactions, identify membrane proteins involved in binding transported effectors, etc. [30].4.Metabolomics*:* Much more remains to be learned about sugar metabolism during infection, and new tools in metabolomics, such as in situ detection of plant lipids and ions, could explore additional processes impacted by *U. maydis* [31,32].

A series of interrelated questions concerns how maize cells contribute to organ formation. Which maize genes are required to redirect cells into the tumor pathway? Are the same genes required in all cell types, tissues, and organs, or are there discrete pathways? Are proliferating cells directed to form tumors by the same steps as non-proliferating cells that were stimulated by See1 or other effectors? How are multiple maize cells coordinated within a tissue or organ to result in a coherent tumor, i.e., with sufficient epidermis to enclose the tumor and a solid mass within the tumor? How is space generated within a growing tumor to accommodate fungal development?

Similarly, it is unclear why tumor formation is so variable within a tissue and between similar stage plants. A wide variety of stochastic processes are involved in infection success including the speed with which cells are infected, the proportion of each cell type that is infected, and the effective fungal dose. It is unsurprising that the combination of many of these stochastic factors results in a broad distribution of disease phenotypes, although it is unclear which of these processes are the main contributors to the observed variability. Cell development within an anther is highly synchronized, meaning that for anthers that are the same size variability between cell type abundance is unlikely to be a key factor [6].

As more details are discovered regarding the maize–*U. maydis* interaction, it is also important to ask why this “pathogen” causes so little damage in typical maize fields. No strong R (resistance) alleles have been identified in maize [20]—has the co-evolution of host and pathogen has resulted in such a weak pathogen that its presence on individual plants or in the population is tolerated with few systemic or reproductive consequences? Tassels are large and designed to shed copious pollen for a week or more—does loss of a subset of anthers really matter? Similarly, does tumor formation by conversion of individual kernel primordia or developing kernels to tumors sufficiently suppress reproductive success to be consequential? Even if the agricultural impact of this host–pathogen interaction is minimal, analysis of maize host responses to *U. maydis* has already uncovered multiple mechanisms of disruption of normal host development and key fungal genes required for infection and tumor progression. Implementation of new techniques and a focus on individual host cell types will undoubtedly enrich our understanding of both fungal strategies to stimulate tumors and host pathways that result in the maize contribution to this novel organ type.

## Figures and Tables

**Figure 1 jof-07-00008-f001:**
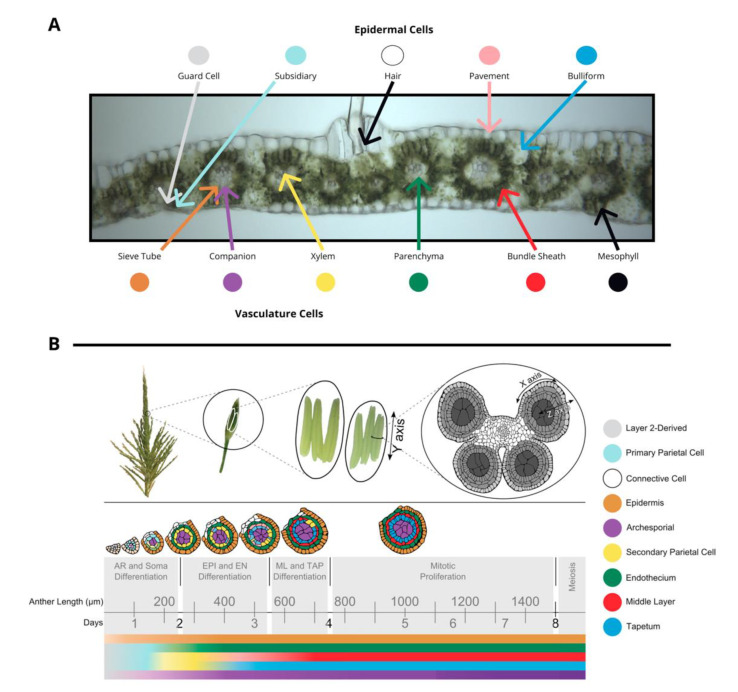
(**A**) A transverse image of a hand section of an adult leaf with different cell types labeled: bundle sheath, mesophyll, epidermal cells (guard cells, subsidiary, hair, pavement, bulliform), and vasculature cells (sieve tube, companion, xylem, parenchyma). (**B**) Tassels contain many paired spikelets with two florets with three anthers each. Anthers are composed of four lobes joined with vasculature as shown in a transverse section. The time course of anther development is segmented into grey boxes on the basis of differentiation and proliferation of key cell types with anther length and time shown for reference. Anther lobes were traced from confocal images with different cell types filled in colors corresponding to the legend. The colored bars correspond to differentiation of different cell types over time. (**A**) was provided by Susanne Matschi and (**B**) is reprinted from [5].

**Table 1 jof-07-00008-t001:** Maize mutants worth exploring.

Disrupted Process	Gene Name	Notes
Photosynthesis	*iojap*	Random loss of chloroplast ribosomes results in stripes of white, yellow, pale green, and normal green on leaves. In this chimera, you can test mutant and impaired tissue simultaneously.
Chloroplast greening	*zb4*	Transverse leaf sectors (physiological chimera likely caused by day or night temperature) allow study of fungal growth in affected and normal areas simultaneously.
Leaf sugar metabolism	*tie dyed 1, 2*	Differentially impacted areas on the same leaf.
Anther developmental timing	*ms8*	Heterochronic anther mutant in which specific steps are delayed; other examples include *csmd1.*
Conditional anther growth	*dcl5, ocl4*	Temperature-dependent phenotypes permit generating a chimeric tassel of normal and arrested anthers by temperature treatment for 1 or a few days.
Response to pathogens	Diverse *les*	Mutants spontaneously express symptoms of specific pathogen infections resulting in zones with or without host leaf responses.
Leaf and stem development	*dwarf, nana*	Multiple loci disrupted in gibberellin (*dwarf*) or brassinosteriod (*nana1*, *2*) biogenesis or response.
Any target		Utilize CRISPR/*Cas9* to disrupt genes of interest.

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
