# Peer review of "Understanding Ustilago maydis Infection of Multiple Maize Organs"

_jof, 2020, doi:10.3390/jof7010008_

Round 1

Reviewer 1 Report

Dear Authors
Please update your review article with more reviews in 2019, 2020, and 2021.
See some examples below:

Xia, W., Yu, X. and Ye, Z., 2020. Smut fungal strategies for the successful infection. Microbial Pathogenesis142, p.104039.

de la Torre, A., Castanheira, S. and Pérez-Martín, J., 2020. Incompatibility between proliferation and plant invasion is mediated by a regulator of appressorium formation in the corn smut fungus Ustilago maydis. Proceedings of the National Academy of Sciences117(48), pp.30599-30609.

Schurack, S., Depotter, J.R., Gupta, D., Thines, M. and Doehlemann, G., 2020. Transcriptome analysis in the maize-Ustilago maydis interaction identifies maize-line-specific activity of fungal effectors. BioRxiv.

Rodríguez Rivas, Á., Ramos Barrales, R. and Pejenaute Ochoa, M.D., Study of PMT target specificity in Ustilago maydis.

Schmitz, L., Kronstad, J.W. and Heimel, K., 2020. Conditional gene expression reveals stage‐specific functions of the unfolded protein response in the Ustilago maydis–maize pathosystem. Molecular plant pathology21(2), pp.258-271.

Depotter, J.R., Zuo, W., Hansen, M., Zhang, B., Xu, M. and Doehlemann, G., 2020. Effectors with different gears: divergence of Ustilago maydis effector genes is associated with their temporal expression pattern during plant infection. bioRxiv.

Vijayakrishnapillai, L.M., Desmarais, J.S., Groeschen, M.N. and Perlin, M.H., 2019. Deletion of ptn1, a PTEN/TEP1 Orthologue, in Ustilago maydis Reduces Pathogenicity and Teliospore Development. Journal of Fungi5(1), p.1.

.

Author Response

We have added more papers (references 11,17,19, and 29) that were published in 2019 or later as suggested by reviewer 1 although we have chosen to only include peer-reviewed articles on effectors (as opposed to other U. maydis virulence associated genes which are beyond the scope of this review).

Reviewer 2 Report

Few is known about the tissue specific interplay between Ustilago maydis effectors and Zea mays. But what is known is nicely summarised in this review. It is clearly structured and very well written. In particular, the review lists methods and strategies to address the question, how particular effectors of the pathogen affect different tissues and organs of the maize in the future.

Typo:

In the legend of Figure 1 (B) "joined with" is doubled.

Author Response

Line 52: we have corrected the typo in the figure legend.